# Promotion of the Application of BIM in China—A BIM-Based Model for Construction Material Recycling

**Kefei Zhang** [1] and **Jing Jia** [2,*]

1   School of Mechanical and Civil Engineering, University of Birmingham, Birmingham B15 2TT, UK; zh_kefei@163.com
2   Department of Civil Engineering, Collage of Engineering, Ocean University of China, Qingdao 266100, China
*   Correspondence: jingjia@ouc.edu.cn; Tel.: +86-137-9194-0195

**Abstract:** The recovery rate of construction materials is only 5% in China, which will lead to environmental and economic problems. Researchers from other countries have recognized the potential of building information modelling (BIM) in optimizing construction material recycling. However, previous research did not take the whole life cycle into consideration and was not practical enough. In this research, a questionnaire was conducted to find out how construction waste is disposed of in construction projects. Then, the existing research results were analyzed to find out how to apply BIM in the whole-life-cycle disposal of construction materials. According to the results of the questionnaire, landfill is the most common way to dispose of construction materials in China; besides this, almost no construction projects use BIM in material recycling. Hence, a BIM-based dynamic recycling model is proposed. Information management of materials, demolition planning, and BIM were all combined in this model for the purpose of optimizing the application of BIM, thus developing a waste material disposal system to achieve higher recovery rates and sustainability. More positive measures should be taken to deal with the problem of construction waste; if not, more environmental and economic problems will follow.

**Keywords:** BIM; recycling; construction material; BIM-based recycling model

## 1. Introduction

The construction industry accounts for a significant percentage of global material consumption [1] and is responsible for material waste [2], which has a major negative impact on the global ecological environment [3]. According to Hebel et al. [4], in 2025, China will produce more than half of the total solid waste generated by cities, mainly in the construction industry, all around the world. Furthermore, from data released by the National Development and Reform Commission of China [5], from 2009 to 2014, only 5% of construction and demolition waste was recycled. The waste that cannot be properly disposed of will cause serious damage to the Earth and environment.

Under these circumstances, there is a great demand for waste recycling of building materials in China; the economic and environmental benefits gained from recycling are enormous [6], and recycling is now seen as a solid waste management strategy [7]. In the last decades, building information modelling (BIM) has received rapidly increasing attention from both academics and the construction industry globally [8]. The application of BIM technology covers many aspects of a building project, including building design, cost estimation, 3D coordination, facility maintenance, and building performance analysis [9]. In some countries, the feasibility of applying BIM in the deconstruction or demolition phase and recycling of construction materials has been proved in many studies [10–14]. According to Spisakova and Mandicak [15], more efficient construction and demolition waste (CDW) management and minimization can be achieved through BIM; besides this, a more specific estimation of recyclable materials can also be realized. However, all the research in this field has been confined to a certain stage, not the whole life cycle of a

construction project. Meanwhile, almost all this research is only conceptual, without any specific implementation plan or framework.

In China, BIM is considered as an essential technology to achieve sustainability of the construction industry [14]. However, there is almost no relevant research about applying BIM in the improvement of construction waste recycling. Under these circumstances, this paper attempts to analyze the development status of BIM, as well as the current situation of waste and recycling of building materials in China, through a questionnaire survey. Then, based on the existing research findings and their limitations, this research aims to further prove the rationality of applying BIM to the recycling of construction materials. Furthermore, the demolition and waste management plan proposed by Ge et al. [10] is also optimized in this dissertation to obtain a more detailed and dynamic recycling model which can cover the whole life cycle of a construction project with a specific application plan that is different from those in other research. Finally, the rationality of this model is proved through the questionnaire.

## 2. Materials and Methods

### 2.1. Structure of the Research Methodology

The research methodology used in this paper is mainly divided into two parts: literature survey and questionnaire survey. The specific logical order is summarized in Figure 1.

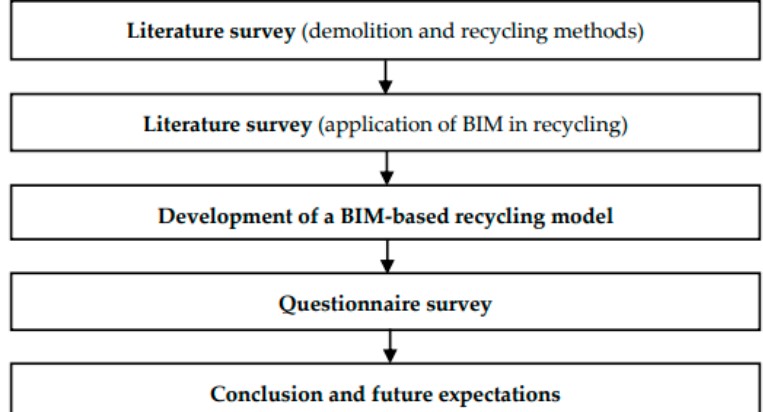

**Figure 1.** Methodology structure.

### 2.2. Selection of the Research Method and Data Collection

The qualitative research method that is used in this paper was chosen as one of the main research methods [16]; compared with other methods, it is faster and more creative [17]. A questionnaire, as one kind of qualitative research method, is a widely used and useful instrument for collecting information, and it is comparatively straightforward to analyze [18]. In this research, we attempt to determine the current situation of the application of BIM in China and develop the existing recycling plan to propose a dynamic and visual conceptual recycling model through the use of BIM. First, analysis of the findings from other research was used to obtain the proposed model. Then, a questionnaire survey was conducted in the form of an online survey to obtain suggestions and opinions on the developed model mentioned above; a questionnaire makes it easy to obtain information from target groups of people at low cost in terms of time and money, while acquiring useful data for testing a hypothesis [19].

### 2.3. Design of the Questionnaire Survey

2.3.1. Structure of the Questionnaire

This questionnaire survey was divided into two parts, as shown in Table 1. First was a survey about the current situation of the BIM adoption status and waste of building materials in China. Participants answered the questions according to their own study

findings and real work experience. The proposed BIM-based recycling model is introduced in the second part, with a clear and succinct explanation attached. Some questions about this model were asked to obtain opinions and advice from the participants.

**Table 1.** Structure of the questionnaire.

| | Section | Main Content | Question Number |
|---|---|---|---|
| | Background of the participants | Occupation and duration | 1,2 |
| **Part 1** | BIM | Application of BIM in China | 3,4,5,6 |
| | Waste and recycling | Waste of building materials in China and situation of recycling | 7,8,9,10,11,12 |
| **part 2** | Proposed BIM-based recycling model | Feasibility of the model and advice on it | 13,14,15,16 |

2.3.2. Selection of the Participants

The target groups in this questionnaire survey were people who engage in research related to BIM or who work in the construction industry and are familiar with BIM. As Table 2 shows, a total of 37 people participated in this questionnaire, of which 35% (13) were academics, and the remaining 24 participants were managers at different levels, from junior manager to senior manager, working in the construction industry in China. It is worth mentioning that more than half of the participants entered the construction industry less than five years previously; these participants are exposed to the latest technical developments and are more receptive to innovation in traditional methods.

**Table 2.** Types and lengths of the occupations of participants.

| Types | Number | Length | Number |
|---|---|---|---|
| Academic | 13 | 1–5 years | 20 |
| Junior Manager | 17 | | |
| Intermediate Manager | 6 | 6–10 years | 14 |
| Senior Manager | 1 | 15+ years | 3 |
| Total | | 37 | |

2.3.3. Ethical Considerations and Limitations of the Questionnaire

There were no confidential questions relating to any companies or organizations in this questionnaire. However, considering the privacy of all participants, this survey was conducted anonymously, and none of the questions involved any personal information. Furthermore, during this research, all participants could obtain access to the data to check the process of the survey and add comments on it.

The shortcomings of this questionnaire are as follows:

1. Uncertainty of the sample size;
2. Negative influence on the quality of responses due to the literacy level of participants;
3. Possible unsophistication of the advice given for the proposed recycling model;
4. The data quality cannot be guaranteed since questionnaires are often completed hastily and carelessly [13].

**3. Research Results and Discussion**

*3.1. Questionnaire Survey Part One: Necessity of Development*

Adoption Status of BIM:

According to the research results, more than 90% of the participants admitted that BIM brings numerous managerial and economic benefits to the construction industry. However, 92% (34/37) of the respondents indicated that the proportion of projects using BIM is less than 30% based on their research and experience. As Cao et al. [20] said, the application or development of BIM in China is still in its infancy.

Moreover, in these projects that used BIM, BIM was mainly used in the conceptual planning and feasibility study phase (54%) and the design and engineering phase (51%). Only 5% of construction projects applied BIM in the demolition phase, let alone adopted it in the recycling of building materials; this does not take full advantage of BIM and should be developed. This result was also confirmed in the research of Cao et al.—that, in China, the application of BIM covers only the design and construction phases of a project [18].

Waste and Recycling of Building Materials:

As shown in Figure 2, 86% [(18 + 14)/37] of the participants agreed that the waste of materials and facilities is severe during the demolition phase. Furthermore, almost 38% (14/37) of the respondents "strongly agree" with this statement. Combining both the findings through literature survey and the questionnaire responses, it is clear that China does face a serious problem in the waste of construction materials at present. It is thus imperative to reform the demolition and recycling system.

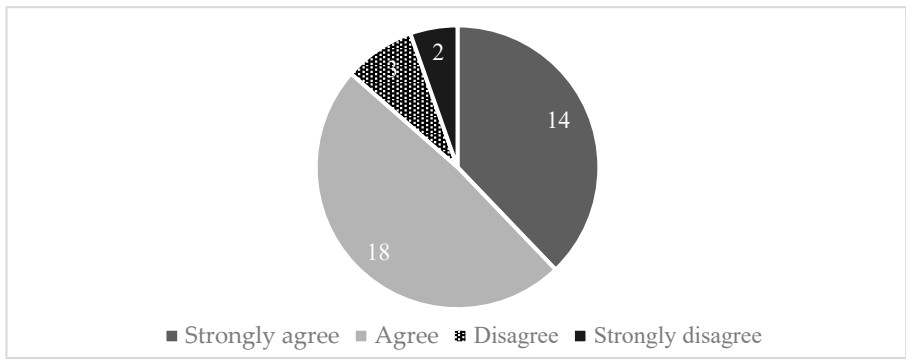

**Figure 2.** Respondents' attitudes regarding the statement that in China, building material waste during the demolition phase is a serious issue.

A main contributor to the serious waste of construction materials is improper disposal of waste materials. More than 50% of respondents (21/37, Figure 3) said that, based on their experience and research, in China, during or after the demolition phase, the most common way to dispose waste materials is landfill; this is in agreement with the results from other research in this field. Even though recycling is the second most common method of disposal, the status quo of a low recycling rate remains.

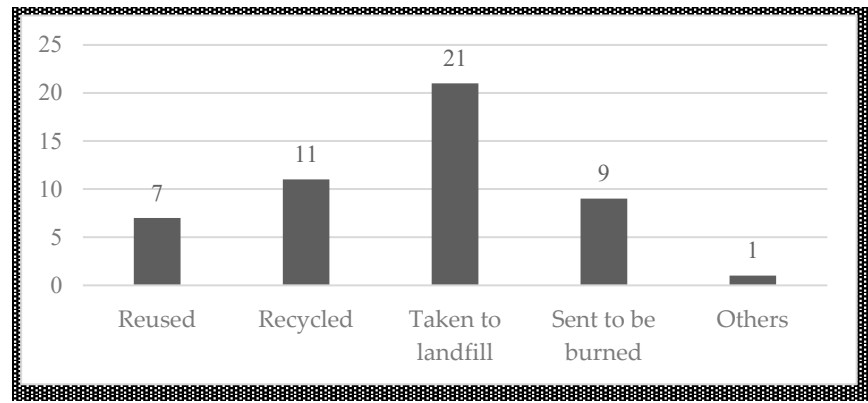

**Figure 3.** Frequency of using different disposal methods of building material waste.

As shown in Figure 3, landfill is the most common method, and the adoption of recycling in the disposal of building materials is relatively low. This also confirms the research result from Luo et al. that in China, the most common method of dealing with waste is to bury it [21]. Then what causes the low recycling rate (5%) of construction materials in China? Participants were asked to rank their degree of agreement or disagreement with the reasons listed in Figure 4. It is noticeable that the top two reasons were

1. Inappropriate demolition and recycling plan (R = 89%; 33/37) and
2. China does not have a mature industrial chain for recycling waste building materials (R = 86%; 32/37). (Here, R is the percentage of respondents who agree or strongly agree with the given reason.)

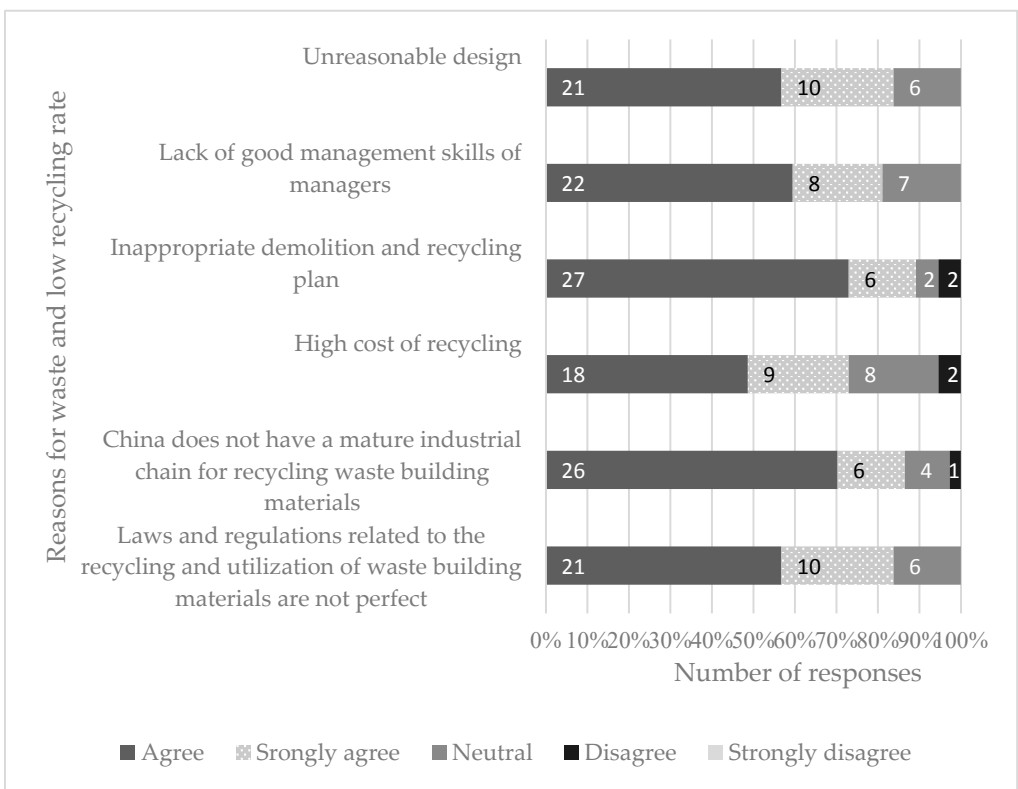

**Figure 4.** Respondents' views on reasons for building material waste and the low recycling rate.

According to these participants, the inefficiency of demolition and recycling plans and the lack of a mature chain for building material recycling lead to the waste of construction materials, with a low recovery rate and pressure from environmental problems on the construction industry in China. This result emphasizes the necessity of redeveloping the traditional demolition and recycling plan.

### 3.2. Proposed Developed BIM-Based Recycling Model

### 3.2.1. Developing the Existing Model

Figure 5 shows the developed deconstruction and waste management process proposed by Ge et al. [10], which is an improvement on the traditional demolition and recycling plan. However, in this model, BIM is only applied to buildings that did not use BIM during construction to classify materials, which is not enough to optimize the recycling plan for construction materials.

The marked parts in Figure 5 the adoption of BIM and the demolition plan, can be improved to be more specific and practical. Through consideration of the application status of BIM in China, together with analysis of the model developed by Ge et al. [10], one optimized model which can cover the whole life cycle of a project is proposed herein.

### 3.2.2. Proposed Model

Through comprehensive consideration of the literature findings and the results of the questionnaire, for the purpose of improving the current situation of serious building material waste and low material recovery rates, one developed model for building material

recycling is proposed as follows. This model combines three dimensions of BIM: 3D modelling, 6D sustainability, and 7D facility management [22].

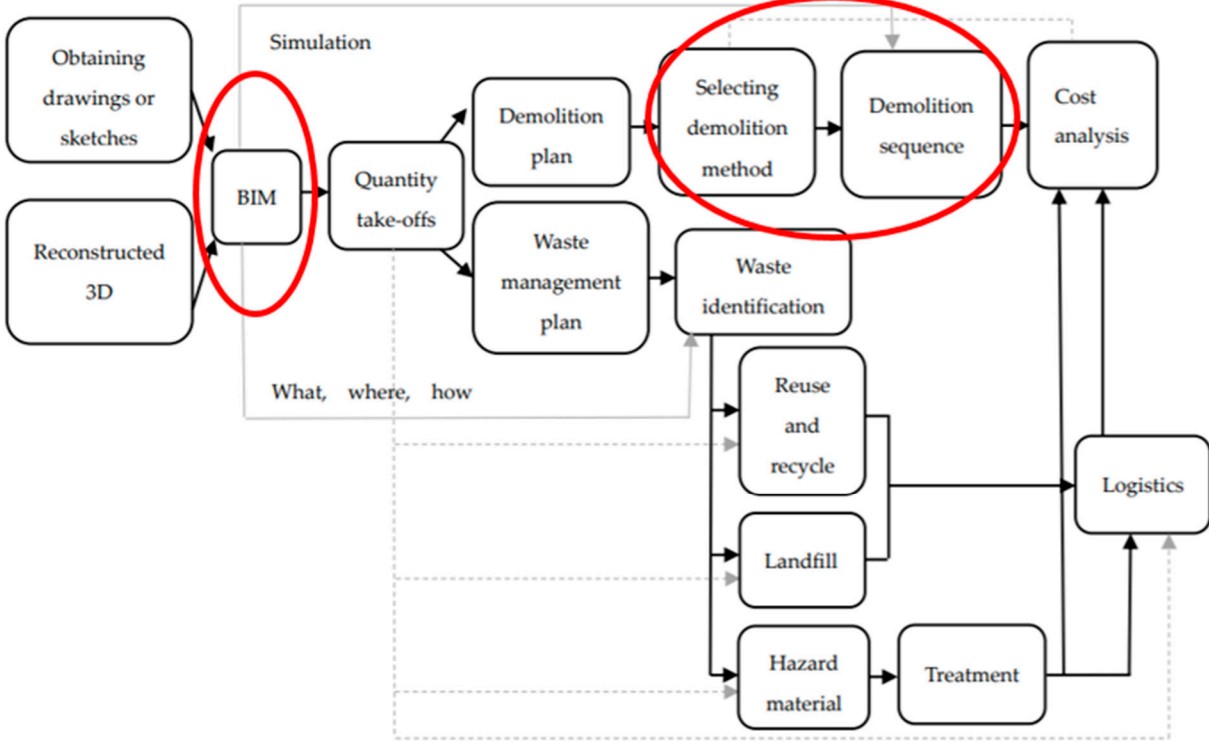

**Figure 5.** Model of BIM application in the demolition phase from Ge et al. [10].

### 3.2.3. Design and Conceptual Phase

First, sustainable design and the use of recyclable materials should be taken into consideration in the conceptual and design phase for the purpose of simplifying the demolition processes and reducing the complexity of material recycling. At the end of this stage, sketches or drawings can be obtained as the basis for building the 3D BIM model in the next stage.

### 3.2.4. Construction and Operation Phase

Then, in the construction and operation phase, a BIM-based model is established; this involves three dimensions (modelling) of the BIM according to the design and drawings gained in the first stage. Furthermore, as the project progresses, any inevitable changes related to building materials, as well as all information (type, quantity, location etc.) on materials used, should be recorded in this 3D BIM model, forming the basis of the material database in the next stage.

### 3.2.5. Demolition Phase

After the construction and operation phase, a final 3D BIM model can be acquired, including a database of the materials used, containing their type, quantity, location, and recyclability. The arrangement of demolition plans, sequence, and scheduling, involving four dimensions (scheduling) of the BIM, can vary with the specific situation of material use. Then, during actual demolition processes, waste materials that have already been obtained can be disposed of through four methods, as shown in Figure 6. Information about the disposed-of materials is updated in the final 3D BIM model. Managers can combine the updated model and current situation to decide whether to change or continue with the original plan. The circulation, marked by bold lines in Figures 5 and 6, is a dynamic and visible process through which managers can obtain accurate information

and then make suitable and economical decisions on the rest of the demolition work. With the help of BIM assisting in the selection of the demolition method and developing the demolition stages [3], more time and financial benefits can be obtained. Prior to all of these, the recovery of construction materials will be developed.

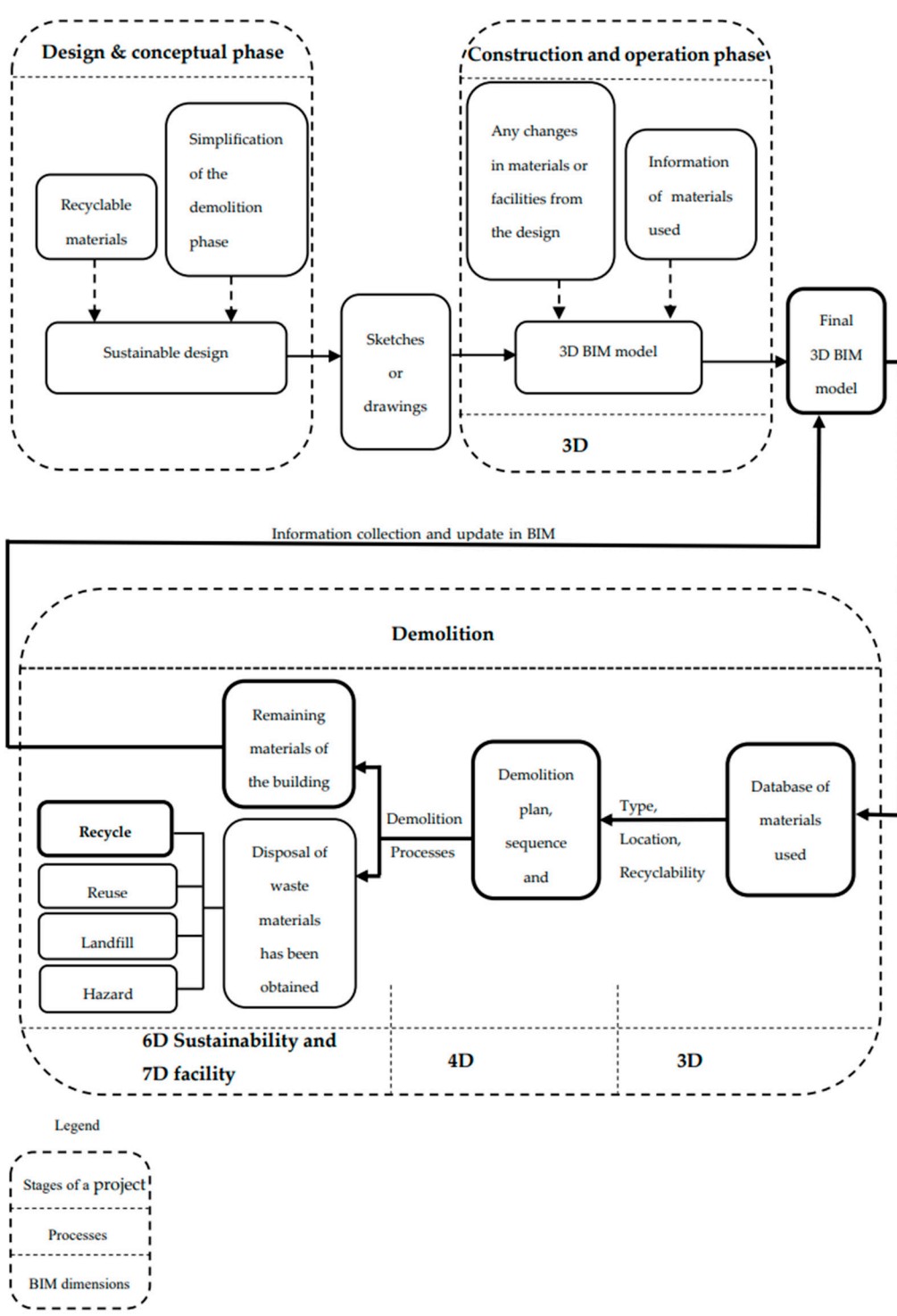

**Figure 6.** The developed recycling and demolition model.

### 3.2.6. Comparison with Results from Other Research

Compared with research from Spisakova and Mandicak in which a more specific estimation of recyclable materials was achieved, the BIM-based model proposed above realizes an accurate estimation of recyclable materials, as well as other materials, through the database of materials used. Besides this, different from the model developed by Ge et al. [10], this model covers the whole life cycle of a construction project, through which information about all construction materials can be tracked. It is also a dynamic model that can maximize the function of BIM in material recycling.

### 3.3. Questionnaire Survey Part Two: Feasibility of the Model and Advice

### 3.3.1. Universality of Applying BIM in Recycling in China

First of all, almost 68% (Figure 7) of the respondents have never heard about adopting BIM in construction material recycling. Furthermore, from the questionnaire results, only 8% of participants had been exposed to projects that implemented BIM in recycling, showing that the application of BIM has not yet reached the demolition phase, especially the recycling of building materials. It is valuable to bridge the gap between China and other countries like Singapore, where the use of BIM for material recycling at the demolition phase is already in place [23].

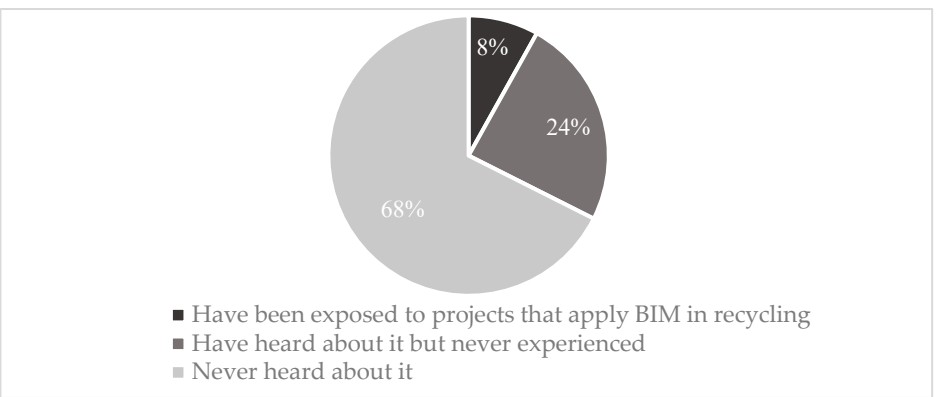

**Figure 7.** Respondents' familiarity with applying BIM in recycling.

### 3.3.2. Feasibility of the Developed Model

In the last three questions in this questionnaire, respondents were asked to give their comments on the proposed recycling model, which had been introduced to them in detail. The results, as demonstrated in Figure 8, show that this model obtained approval from 92% ((21 + 13)/37) of the participants. Among these 34 people, 62% (21/34) thought that this model is feasible in real construction projects and will improve the present situation of building material recycling in China, while in the opinion of the other 38% of respondents, this is a viable model but still needs to be improved.

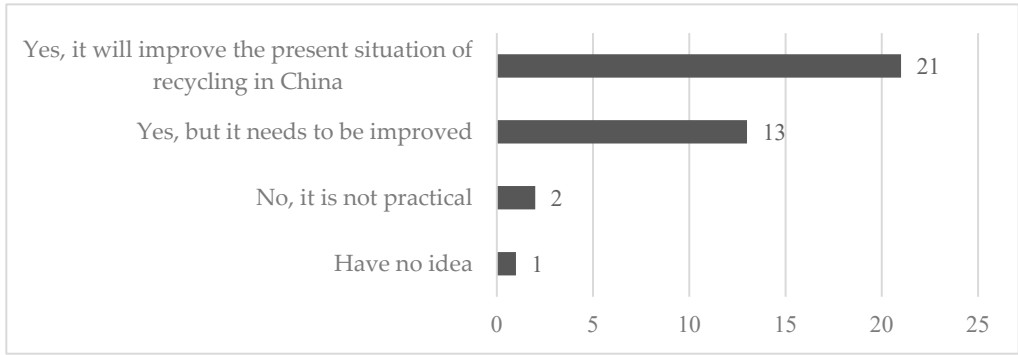

**Figure 8.** Respondents' views on the feasibility of the proposed model.

At the same time, participants were asked, if they were project managers, would they apply this model to their projects to reduce material waste and increase the recovery rate, even though it may mean more cost? The result is that, nearly 95% (35/37, Figure 9) of the participants gave positive responses, that they would use it or would give consideration to it. These responses, given based on the research or work experience of all participants, show excellent applicability of this model in the construction industry of China.

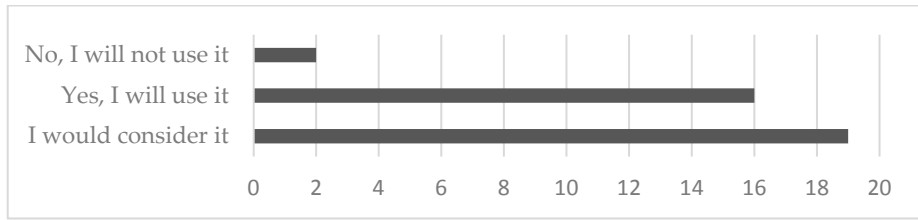

**Figure 9.** Respondents' degree of acceptance of the proposed model.

However, there were still three participants who gave negative evaluations of this model, with two participants thinking it is not practical and one participant having no idea. In addition, there were two participants who responded that as a project manager, they would choose not to use this model. The reasons they gave can be summarized as follows:

1.  Communication for information is more important than a dynamic visualized model for reducing waste and increasing the recycling rate.
2.  Reducing waste is more important than recycling it.

In addition, some suggestions and expectations were given for this developed model:

*"This model can be further improved to reduce waste through making the best use of BIM at design phase."*

*"Organization management in construction can be combined to this model to reduce waste and increase the recovery rate at construction phase."*

*"Hoping the application of BIM in recycling can be popularized in China to improve the current situation."*

*"China should promulgate relevant laws and regulations."*

## 4. Conclusions and Future Prospects

### 4.1. Conclusions

In this research, we investigated the application of BIM to construction material recycling to reduce waste and increase the material recovery rate in China. For this purpose, a BIM-based developed recycling model that can follow the whole life cycle of a construction project was proposed, based on a discussion on the current situation of the application status of BIM and the low recovery rate of construction materials in the construction industry in China. Furthermore, the feasibility was verified through a questionnaire among people who are engaged in work or research in the construction industry in China. The investigation proved that this proposed model is rational and practical given the current inability to properly handle construction waste in the construction industry. More intelligent and environmentally friendly construction processes can be achieved through this model, as well as less waste and a higher recycling rate.

Based on our research, the following conclusions can be drawn:

1.  Inefficiency of demolition and recycling plans and the lack of a mature chain for building material recycling lead to the waste of construction materials;
2.  The most common method at present for the disposal of waste materials is landfill;
3.  The application of BIM in recycling in China and the proposed model in particular were supported by more than 90% of participants;

4.  Development of the current recycling plan is imperative, and the proposed model will provide an improvement in material recycling in China, with economic and environmental benefits.

### 4.2. Limitations

This research achieved the expected goal; however, there are still some ineluctable limitations. First of all, it is obvious that, limited by time and distance, the sample size of the questionnaire survey is small, which may lead to results that are not sufficiently convincing. Further, even though this proposed model gained support from most of the respondents, theory and practice are inseparable. A questionnaire is not as good as applying the model to real construction projects, which is a more convincing to verify the practicability of it. However, because of time and technology limitations, the feasibility could only be testified through a questionnaire survey.

### 4.3. Recommendations and Future Prospects

The positive effect of BIM on reducing material waste and improving recovery efficiency is noticeable. At present, the most important thing is using it in real projects and carrying out further research to form a mature BIM-based industrial chain of recycling materials. More money and resources should be spent on developing the application of BIM, reducing waste, and recycling. People working in the construction industry should be educated about emerging technologies and be more accepting of new methods of waste material recovery. Besides this, according to a senior manager in the questionnaire survey, all these new methods or systems of material recycling will not work well without binding laws and regulations. Hence, to completely solve the problem of waste and low recycling rates, the government of China should put tough measures in place to urge the construction industry to reform.

**Author Contributions:** K.Z. contributed to the conception of the study and performed the data analysis as well as the preparation of the manuscript. In addition, K.Z. also designed the questionnaire and organized the questionnaire survey. J.J. performed the data analysis with constructive suggestions. Both authors have read and agreed to the published version of the manuscript.

**Funding:** This research was funded by the Natural Science Foundation of China (NSFC), No.51908523, to whom we are grateful.

**Institutional Review Board Statement:** Not applicable.

**Informed Consent Statement:** Not applicable.

**Data Availability Statement:** Data available in a publicly accessible repository.

**Conflicts of Interest:** The authors declare no conflict of interest.

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
