# Peer review of "Promotion of the Application of BIM in China—A BIM-Based Model for Construction Material Recycling"

_recycling, doi:10.3390/recycling6010016_

Round 1

Reviewer 1 Report

Please explain the acronym BIM, because many researchers interested in construction and demolition waste dont know the building information modeling

Does the BIM-based model for construction materials recycling take human factors into account?

explain if In the future will be necessary standards of the work for the selection of demolition method or sample work to check the quality

Author Response

Dear Editor,

Thank you for your comments and suggestions, and thank you very much for your time that you spend on our manuscript. The answers to specific questions are as follows.

Reviewer 2 Report

The article aims to understand the 'potential of BIM in optimizing construction materials recycling'. To this end, it uses surveys which, as a first stage, assess the use of BIM in the design phase. Secondly, they evaluate the use of a model that optimizes the use of BIM for the management of construction and demolition waste considering the entire life cycle of materials.

The survey used is divided into 2 parts, the first of which has 12 questions and the second 4. Thirty-seven people were surveyed, of these 30 are academics and junior manager. 20 have between 1 and 20 years of experience. At no time does the target population or the sample size, nor the way in which the professionals who respond to the surveys were chosen, as well as the reason for managers and academics.

Based on the results of the surveys, the low use of BIM in projects in China is mentioned, as well as the lack of knowledge about the possibility of use in construction and demolition waste management.

In general, the article is based on a small set of responses to surveys, without reference to other sources that could strengthen the information. The small number is added to the experience and background of the respondents, with a diversity that gives rise to doubts about opinions. It also asks that they comment on specific aspects, without leaving room for other opinions.

The references should be improved.

Specific aspects:

- Line 57 - review 'which will be used in this dissertation';

- Line 98 – review this paragraph 'In the research or work experience of nearly 92% (34/37) respondents, less than 30% construction 98 projects they experienced has used BIM among which less than 10% projects used BIM in the 99 experience of almost 65% (24/37) participants albeit 92% (34/37) admit that BIM did bring...' It is not clear.

- Line 106 - review 'Almost no construction projects in China apply BIM at demolition phase let alone adopt it in the recycling of building materials' – based on 37 surveys?

- Line 103 (Figure 2) - I do not understand the data in figure, refers to the percentage of projects that used BIM, then numbers appear inside without any reference to their meaning.

- Line 117 (Figure 3) - In this case the legend of the Figure is incomplete, as well as a numbering that is not understood again.

- Line 125 - 'As shown in Figure 3.4' - Check the numbering of the Figure. In this same line it refers '... notwithstanding landfill is the most common method, the adoption of recycling in disposal of building materials is not in the minority.' But does it conclude only based on the surveys? Is there no other data that can confirm this information?

- Line 143 - the section starts with a Figure; it is recommended to insert text that contextualizes it.

- Line 148 – It states that ‘…BIM is just applied in buildings that did not use BIM during construction to classify materials which is not enough to optimize the recycling plan for construction materials.’ But I’m not sure that this is what the Figure shows…According to the model there are two paths, but ‘quantities take-offs’ after BIM application are used (and materials need to be classified).

- Line 165 (Figure 8) - correct ‘Demolitio’

- Line 185 – correct ‘The circulation, marked by bold lines in Figure 3 .7’ – the number is not right.

- Line 196 – ‘It is valuable to bridge the gap between China and other countries mentioned at 196 chapter 2 in the adoption of BIM i n recycling.’ What chapter?

- Line 199 - Figure 9 also needs to be clarified, as well as reviewing its title.

- Line 203 – Please correct ‘introduced to them in detail. The results, as demonstrated in Figure 3 .9…’

- Line 209 – Please correct ‘Surprisingly, nearly 95% (35/37 , Figure 3.10 )…’

Author Response

Thank you for your comments and micromesh suggestions. And thank you very much for the time you spend on our manuscript.

Reviewer 3 Report

  1. Too many grammatical errors within the paper. I suggest you send the paper to a professional English editing services. A few have been corrected below:

‘The construction industry…’ (L.27)

‘impact on the global ecological environment…’ (L.28)

‘will inflict serious damage to the earth and environment.’ (L.33)

‘achieve sustainability within the construction industry’ (L. 38)

‘as shown in Table 1’ (L.67)

‘As shown in Figure 2’ (L.76)

  1. Space missing between;

 ‘research’ and the reference numbers (L. 37)

‘BIM’ and ‘As Figure 2’ (L.76)

‘Figure’ and ‘9’ (L. 192)

  1. The introduction is too short, please provide further background on the use of BIM, you mentioned the research conducted is only conceptual but it’s important you analyse those investigations (by referencing to the work) and justify why your research is novel.
  2. To many referencing errors, for example:

Ge [8] is supposed to be Ge et al. [8]

Hebel [4] is supposed to be Hebel et al. [4]

  1. Please improve the quality of fig. 1
  2. Modify the layout of the paper according to the journal requirements (section and subsection font and numbering)
  3. Please re-write L. 98 to L. 101 (missing punctuations)
  4. The draft is really weak in terms of graphical works. Moreover, all figure captions are short and unclear. More professional tables and figures would be expected. Please improve all figures and tables, carefully and comprehensively, for the quality, quantity, and content. (add colour, use axis titles)
  5. Please improve the conclusion
  6. Please add existing results from similar research to provide a comparison of results or methods.
  7. ‘mentioned at chapter 2’ (L. 197) what do you mean by chapter 2, please re-phrase

Author Response

Thank you for your comments and suggestions. And thank you very much for your time that you spend on our manuscript.

Round 2

Reviewer 2 Report

I have no more comments, the authors answered and followed my suggestions.

Reviewer 3 Report

This paper is ready for publication